# Antioxidant and Cytoprotective Properties of Cyanobacteria: Potential for Biotechnological Applications

**DOI:** 10.3390/toxins12090548

**Published:** 2020-08-26

**Authors:** Adriana Guerreiro, Mariana A. Andrade, Carina Menezes, Fernanda Vilarinho, Elsa Dias

**Affiliations:** 1Laboratory of Biology and Ecotoxicology, Department Environmental Health, National Institute of Health Dr. Ricardo Jorge, Avenida Padre Cruz, 1649-016 Lisboa, Portugal; adriana.guerreiro@insa.min-saude.pt (A.G.); carina.menezes@insa.min-saude.pt (C.M.); 2Faculty of Sciences, University of Lisbon, Campo Grande, 1749-016 Lisboa, Portugal; 3Laboratory of Chemisty, Department Food and Nutrition, National Institute of Health Dr. Ricardo Jorge, Avenida Padre Cruz, 1649-016 Lisboa, Portugal; mariana.andrade@insa.min-saude.pt (M.A.A.); fernanda.vilarinho@insa.min-saude.pt (F.V.); 4Instituto Superior Técnico, University of Lisbon, Av. Rovisco Pais, 1, 1049-001 Lisboa, Portugal; 5Centro de Estudos de Ciência Animal (CECA/ICETA), University of Porto, Rua D. Manuel II, Apartado 55412, 4051-401 Porto, Portugal

**Keywords:** cyanobacteria, antioxidant potential, natural antioxidants, DPPH assay, β-carotene bleaching assay, phenolic, flavonoids, HEK293T cell line, cytotoxicity prevention

## Abstract

Antioxidant compounds from cyanobacteria may constitute a natural alternative to current synthetic antioxidants, which contain preservatives and suspected toxicity. In this work, we evaluate the antioxidant potential of cyanobacterial strains of distinct species/genus isolated from freshwater (n = 6), soil (n = 1) and wastewater (n = 1) environments. Lyophilized biomass obtained from in-vitro cultures of those strains was extracted with ethanol and methanol. The antioxidant potential was evaluated by chemical (DPPH scavenging method, β-carotene bleaching assay, determination of total phenolic and total flavonoid compounds) and biological (H_2_O_2_-exposed HEK293T cell line model) approach. Some strains showed high yields of antioxidant activity by the DPPH assay (up to 10.7% IP/20.7 TE μg/mL) and by the β-carotene bleaching assay (up to 828.94 AAC), as well as significant content in phenolic (123.16 mg EAG/g DW) and flavonoid (900.60 mg EQR/g DW) compounds. Normalization of data in a “per cell” or “per cell volume” base might facilitate the comparison between strains. Additionally, most of the cyanobacterial extracts conferred some degree of protection to HEK293T cells against the H_2_O_2_-induced cytotoxicity. Freshwater *Aphanizomenon gracile* (LMECYA 009) and *Aphanizomenon flos-aquae* (LMECYA 088), terrestrial *Nostoc* (LMECYA 291) and wastewater *Planktothrix mougeotii* (LEGE 06224) seem to be promising strains for further investigation on cyanobacteria antioxidant potential.

## 1. Introduction

Cyanobacteria are a diverse group of photosynthetic prokaryotes, with an estimated number of 8000 species distributed by 150 genera, that, in turn, colonize a variety of aquatic and terrestrial environments, from temperate to tropical and polar regions worldwide [1,2]. Along their evolutionary path, cyanobacteria have developed a multitude of strategies enabling them to adapt and survive in such distinct environments. For example, many cyanobacterial species are predominant in habitats exposed to high solar irradiance and, consequently, developed several mechanisms to protect themselves against the noxious effects of UV light [3]. One of those mechanisms is the production of primary sunscreens such as the pigment Scytonemin or the mycosporine-like amino acids (MAAs) that absorb harmful radiation [3,4]. Besides, MAAs can also act as radical scavengers and protect cyanobacteria from UV-induced oxidative stress [4]. On the other hand, cyanobacteria are an important oxygen source and, as such, must defend themselves against their own induced oxidative environment. Consequently, cyanobacteria produce a variety of secondary metabolites that are effective against reactive oxygen species (ROS), particularly pigments such as carotenoids, as well as polyphenols such as phenolic acids and flavonoid compounds [5,6,7].

Cyanobacteria are a source of bioactive substances with many potential biotechnological applications and their antioxidant compounds, particularly, have been considered promising molecules for the cosmetics [8] and food [9] industries. Indeed, the search for natural antioxidant compounds has gained increasing interest, considering current synthetic antioxidant compounds contain preservatives and suspected toxicity [10]. The marine environment has been a privileged niche for the research of bioactive compounds in cyanobacteria [11]. However, we can currently find in the literature a huge diversity of publications on the antioxidant activity of cyanobacteria from freshwater and terrestrial habitats [5,6,7,9,12,13,14,15,16,17,18]. In general, these studies point to an interest in exploring the antioxidant activity of cyanobacteria, considering their richness of compounds such as carotenoids and polyphenols. Inclusively, some of these authors reported antioxidant activity [5,13,16,17,19,20] and/or antioxidant contents [11,14,15,21] in cyanobacteria quite comparable, and sometimes higher, than those found in eukaryotic microalgae, macroalgae or higher plants [22,23,24,25]. 

The present work aimed to evaluate the antioxidant potential of several cyanobacteria species isolated from freshwater, wastewater and soil. To achieve that purpose, we used both chemical (DPPH inhibition assay, β-carotene bleaching assay, determination of total phenolic and flavonoid compounds) and biological (evaluation of the protective effect of cyanobacterial extracts against H_2_O_2_-induced cytotoxicity in HEK293T cell line) approaches.

## 2. Results

### 2.1. Antioxidant Profile of Cyanobacterial Strains

The results of the DPPH inhibition assay, expressed as the inhibition percentage or as Trolox equivalents (TE), are presented in Figure 1. The methanolic extracts exhibited higher values of antioxidant capacity than the ethanolic extracts, irrespectively of the cyanobacterial strain. The exception was *Planktothrix mougeotii* (LEGE 06224), which showed similar values of antioxidant activity with the two extraction solvents. The methanolic extracts with the highest antioxidant activity were obtained from *Microcystis aeruginosa* (LMECYA 127), *Dolichospermum flos-aquae* (LMECYA 180), *Planktothrix agardhii* (LMECYA 257) and *Planktothrix mougeotii* (LEGE 06224). In those extracts, the DPPH inhibition percentage varied between 8.8% and 10.7%, corresponding to 17.3 TE μg/mL to 20.7 TE μg/mL.

The results for the β-carotene bleaching test are shown in Figure 2. In this case, the ethanolic extracts showed higher antioxidant activity than the methanolic extracts in most of the cyanobacterial strains. *Aphanizomenon flos-aquae* (LMECYA 088) and *Aphanizomenon gracile* (LMECYA 009) exhibited the highest values of antioxidant activity coefficient (AAC) (825.9 and 690.5 AAC, respectively), followed by *Dolichospermum flos-aquae* (LMECYA 180) (384 AAC). On the contrary, methanolic extract of *Microcystis aeruginosa* (LMECYA 127) showed considerably higher activity (456.2 AAC) than the ethanolic extract of the same strain (79.4 AAC). In the remaining strains, the differences between the solvents were not so pronounced.

The content of total phenolic compounds was also higher in the majority of the ethanolic extracts (Figure 3). In those extracts, the highest phenolic concentrations (expressed as Galic Acid Equivalents, GAE) were observed in *Aphanizomenon flos-aquae* LMECYA 088 (123.2 mg GAE/g DW), *Aphanizomenon gracile* LMECYA 009 (102.3 mg GAE/g DW), *Dolichospermum flos-aquae* LMECYA 180 (63.7 mg GAE/g DW) and *Planktothrix mougeotii* LEGE 06224 (61.1 mg GAE/g DW). As for the previous assay, the higher content of phenolic compounds in *Microcystis aeruginosa* LMECYA 127 (67.1 mg GAE/g DW) was detected in the methanolic extract, a similar value to that obtained with methanolic extract of *Aphanizomenon gracile* LMECYA 009 (68.1 mg GAE/g DW). The remaining strains had lower results with both solvents.

As for the previous assays, the content of total flavonoids (Figure 4) was higher in the ethanolic extracts of *Aphanizomenon gracile* (LMECYA 009) and *Aphanizomenon flos-aquae* (LMECYA 088) (606 and 901 mg Quercitine Equivalents―QE/g DW, respectively). The remaining strains showed slightly higher results in extracts prepared with methanol, but the concentration of flavonoids was considerably lower than in the extracts of those two *Apanizomenon* strains.

To refine the comparison between cyanobacterial strains, the antioxidant content per cell and per cell volume (µm^3^) for each strain was determined (Figure 5 and Appendix A). *Nostoc* sp. (LMECYA 291) was the strain with the highest antioxidant content, irrespectively of the antioxidant endpoint. As shown in Figure 5, the antioxidant content per cell (or per cell volume) in this strain was one to three orders of magnitude higher than other strains. *Planktothrix mougeotii* (LEGE 06224), *Microcystis aeruginosa* (LMECYA 127), *Dolichospermum flos-aquae* (LMECYA 180) and *Planktothrix agardhii* (LMECYA 257) followed as the strains with higher antioxidant content per cell. 

### 2.2. Protective Effect of Cyanobacterial Extracts on HEK293T Cell Line against H_2_O_2_-Induced Cytotoxicity

Figure 6 shows the viability of HEK293T cells exposed to a short period (1 h) to cyanobacterial ethanolic extracts followed by a long exposure period (23 h) to H_2_O_2_ (0.1 mM). Data shows that there are no significant differences between the viability of cells exposed to cyanobacterial extracts and H_2_O_2_ (with or without an extract washing step before adding H_2_O_2_) and cells exposed only to the cyanobacterial extracts, for the majority of the strains. It seems, thus, that the extracts somehow reverse the cytotoxic effect of H_2_O_2_. Exception was LMECYA 009 and LMECYA 088, where the viability of cells exposed to H_2_O_2_ after removing the cyanobacterial extracts (Figure 6, top) was lower than the cells co-incubated with extracts and H_2_O_2_ or incubated only with extracts. Even though, the cell viability was higher in cells pre-treated with cyanobacterial extracts when compared with the positive controls (see Material and Methods, item 5.5.1). Indeed, except for LMECYA 009 and LMECYA 291, all the extracts induced cell survival with survival percentages between 7% and 52% (Figure 6, bottom). These were higher when the cyanobacterial extract was washed prior to H_2_O_2_ exposure.

The results were somehow different when cells were exposed for a long period to cyanobacterial extracts (23 H) and a short period (3 h) to a 10× higher H_2_O_2_ concentration (1 mM). The viability of cells pre-treated with cyanobacterial extracts (with or without a wash step) decreased between 43% and 65% (Figure 7, top) after H_2_O_2_ exposure when compared with cells exposed only to cyanobacterial extracts. In general, cell survival (Figure 7, bottom) varied between 5% and 30%, and the inclusion of an extract-wash step before H_2_O_2_ exposure yielded the highest percentages. 

Interestingly, many extracts showed more than 25% of survival in the short extract/long H_2_O_2_ exposure. Extracts from strains LMECYA 173, LMECYA 257, LMECYA 291 and LEGE 06224 were the most effective. In the case of long extract/short H_2_O_2_ exposure, only extracts from strains LMECYA 173 and LMECYA 291 presented a survival rate above 25%. 

## 3. Discussion

This work aimed to evaluate the antioxidant profile of cyanobacterial strains of different species, isolated from freshwaters, wastewater and soil. Antioxidant profiles varied with strain and extraction solvent.

According to the DPPH inhibition assay, methanolic extracts of all strains presented higher antioxidant activity than the ethanolic extracts (except for LEGE 06224, which showed similar results with both extracts). It has been described that methanol is more effective than ethanol in extracting antioxidant compounds [5,17,26], probably due to its high polarity [16]. However, ethanolic extracts presented higher yields for most of the strains by the β-carotene bleaching, total phenolic and total flavonoid assays. This is not surprising since the DPPH assay measures the overall antioxidant capacity of a mixture [26], whereas the other assays are more specific for certain compounds. Besides, the effectiveness of the extraction solvents depends on the nature of the compounds to be extracted. Nevertheless, we should bear in mind that methanol might not be a proper solvent for certain biotechnological applications of cyanobacterial antioxidants. For example, Food and Drug Administration excludes methanol, but not ethanol, from the GRAS (Generally Recognized as Safe) list for food/food packaging industry, due to its recognized toxicity to humans [27]. 

The extracts (methanolic) from *M. aeruginosa* (LMECYA 127), *Leptolyngbya* sp. (LMECYA 173), *Dolichospermum flos-aquae* (LMECYA 180), *Planktothrix agardhii* (LMECYA 257) and *Planktothrix mougeotii* (LEGE 06224) showed the higher yields of antioxidant activity by the DPPH assay with values ranging from 8.8% IP (17.3 TE μg/mL) to 10.7% IP (20.7 TE μg/mL). These values are similar to those obtained with methanolic extracts of *Leptolyngbya protospira* (7.65%), *Nodularia spumigena* (13.02%) e *Phormidiochaete* sp. (14.59%) [5], *Cylindrospermum* sp. (7%) [20], *Plectonema boryanum* (8.5%) and *Anabaena doliolum* (11.3%) [19]. However, higher values were reported in *Nostoc* sp., (28–100%) [5,13,20], in *Oscillatoria agardhii* (51–92%) and *Anabaena sphaerica* (39–62%) [16] and in *Phormidium fragile* (25.5%), *Lyngbya limnetica* (34.5%), *Scytonema bohnerii* (30.8%) and *Calothrix fusca* (21.3%) [17].

Our results of DPPH inhibition assay are quite comparable with previous results obtained with methanolic extracts (1 mg DW/mL) of brown algae (*Bifucaria bifurcata* and *Fucus spiralis*) and green algae (*Enteromorpha intestinalis* and *Ulva rigida*), that varied between 10% and 15% [24]. On the other hand, Martins et al. [28] reported DPPH inhibition activity ranging from 11% and 80% in methanolic extracts (1 mg/mL) of 26 macroalgae species. As for macroalgae, results from DPPH assay in cyanobacteria seems to strongly depend on the species.

Indeed, species-specific results were also reported for other antioxidant endpoints, including pigments such as carotenes [29,30]. However, other authors highlighted that strains of the same genus may have a similar pattern of pigments [31]. In fact, in our study, the extracts (ethanolic) of *Aphanizomenon gracile* (LMECYA 009) and *Aphanizomenon flos-aquae* (LMECYA 088) showed significantly higher values in the β-carotene bleaching assay (690.47 and 828.94 AAC respectively) than the other tested strains. It is difficult to compare these results with others obtained with cyanobacteria because they are expressed in other units. However, our results are within the range [600–5000 AAC] of values reported for ethanolic extracts (1 mg/mL) of several eukaryotic microalgae (*Porphyridium cruentum*, *Phaeodactylum tricornutum* and *Chlorella vulgaris*) [32]. More concentrated extracts of those strains, particularly *C. vulgaris*, produce even higher values (up to 22,000 AAC) and the authors considered that those results were the highest activity ever registered with biological extracts by the β-carotene bleaching assay. 

Some authors have studied the carotenoid profile of cyanobacteria and quantified the individual compounds. Lopes et al. [29], for example, investigated several species from different habitats and found that the terrestrial *Nodosilinea (Leptolyngbya) antarctica* (LEGE13457) and the freshwater *Cyanobium gracile* LEGE12431 presented the highest content in carotenoids, namely of β-carotene (27.7 µg/g and 24.0 µg/g, respectively). Another study with cyanobacteria from rice paddies, showed β-carotene levels ranging from 1221 µg/g (*Nostoc calcicole*; soil) to 8132 µg/g (*Anabaena vaginicola*; water) [21]. These studies demonstrate that both terrestrial and freshwater cyanobacteria might be important natural sources of carotenoids. Indeed, as noted by Hashtroudi et al. [21], carrot juice, pumpkin and cooked sweet potato are the three foods with the highest content of β-carotene, 93, 69 and 115 µg/g, respectively, according to the U.S. Department of Agriculture (USDA-SR24). The same authors also highlight that the highest content of β-carotene has been reported for the commercialized microalgae, *Dunaliella salina* and *Dunaliella bardawil* (11,000–21,000 µg/g dw) [21].

In our study, the ethanolic extracts also showed the highest values of phenolic compounds, which varied between 34.2 and 123 mg EAG/g of DW. *Aphanizomenon gracile* (LMECYA 009) and *Aphanizomenon flos-aquae* (LMECYA 088) exhibited the highest values (102.32 and 123.16 mg EAG/g DW, respectively). These results are similar and/or higher than those observed in other studies. Bavini [12] reported a total phenolic content of 5.4 to 28 mg EAG/mg of extract (10 mg/mL, ethanolic extraction) in eight strains of marine and freshwater cyanobacteria. The higher content was observed in *Synechocystis salina* (marine) and *Phormidium* sp. (freshwater). Yasin et al. [11] described a range of total phenolic compounds from 15.1 to 96.7 mg EAG/mg, depending on the solvent, in the freshwater *Nostoc muscorum* NCCU-442. In a study from Singh et al. [6] with twenty terrestrial cyanobacterial strains, total phenolic content (methanolic extracts) varied between 22 mg EAG/g (*Aulosira fertilissima*) and 290 mg EAG/g (*Oscillatoria acuta*). However, a direct comparison cannot be done because these authors extracted cyanobacteria biomass with methanol, and they did not specify the extract concentration. Also, Rajishamol et al. [15] evaluated the total phenolic content of methanolic extracts (100 μg/mL) of the freshwater *Oscillatoria limosa* (33.4 mg EAG/g), *Synechococcus elongatus* (21.33 mg EAG/g) and *Synechocystis aquatilis* (15.94 mg EAG/g). Aydaş et al. [33] reported a maximum of 78.1 μg EAG/mg in *Synechocystis* sp. BASO673 (methanolic extract, 0.1 g/mL). El-Aty et al. [16] registered 14.8 mg EAG/g and 20.9 mg EAG/g in methanolic extracts of *Anabaena sphaerica* and *Oscillatoria agardhii*, respectively, but the authors did not indicate the extract concentration. The total phenolic content in strains of *Dolichospermum flos aquae* (4.38 mg EAG/g) and *Nostoc ellipsosporum* (39.9 mg EAG/g) were reported by Li et al. [18], but in this study other solvents (water, hexane, ethyl acetate) were used and the concentration of the extract was not mentioned in the paper. Hossain et al. [14] reported lower total phenolic content in aqueous extracts of *Oscillatoria* sp. (2.96 mg EAG/g), *Lyngbya* sp. (5.02 mg EAG/g), *Microcystis* sp. (2.65 mg EAG/g) and *Spirulina* sp. (1.78 EAG/g). When comparing these data with other natural sources of antioxidants, the potential of cyanobacteria becomes obvious. In brown algae (*Bifucaria bifurcata* and *Fucus spiralis*) for example, the total phenolic content varied between 1 mg EAG/g and 29.8 mg EAG/g, depending on the extraction solvent [24]. It is also interesting to compare cyanobacterial results with green tea, which is considered a powerful natural antioxidant, due to its high content of gallic acid, among other compounds [25]. A study on the applicability of green tea in antioxidant-active food packaging showed a variation of total phenolic content between 272 mg EAG/g and 416 mg EAG/g, in several green tea samples [25].

In the present study, ethanolic extracts of *Aphanizomenon gracile* (LMECYA 009) and *Aphanizomenon flos-aquae* (LMECYA 088) also showed significantly higher values of flavonoid content (605.58 and 900.60 mg EQR/g DW, respectively). In the other tested strains, the highest values were observed in methanolic extracts and varied between 127.8 and 371.3 mg EQR/g DW. These results are higher than those obtained in some previous studies with cyanobacteria. Singh et al. [6], for example, reported total flavonoid content 1000 times lower in methanolic extracts of 15 terrestrial cyanobacteria (53–634 μg EQR/g). In methanolic extracts of freshwater species, El-Aty et al. [16] reported values of 3.54 mg EQR/g DW (*Anabaena sphaerica*) and 12.11 mg EQR/g DW (*Oscillatoria agardhii*). Hossain et al. [14], on the other hand, described similar results in aqueous extracts of freshwater *Oscillatoria* sp. (552.59 mg EQR/g), *Lyngbya* sp. (664.07 mg EQR/g), *Microcystis* sp. (392.00 mg EQR/g) and *Spirulina* sp. (483.33 mg EQR/g). Lower total flavonoid content was also reported in macroalgae, such as the study of Osuna-Ruiz et al. [22] with methanolic extracts of *Caulerpa sertularioides*, *Codium isabelae*, *Gracilaria vermiculophylla*, *Padina durvillaei*, *Rhizoclonium riparium*, *Spyridia filamentosa* and *Ulva expansa* (10 to 14 mg EQR/g dw). Raja et al. [23] described concentrations between 8 and 55 mg EQR/g DW in methanolic extracts (2 mg/mL) of *Codium fragile*, *Ulva lactuca* and *Eisenia arborea*.

A comparison of the results among studies might be difficult given the differences in the endpoints and methods used to access antioxidant properties. Besides, differences among strains in the same study should also be considered. In our work, the tested strains have very distinct morphologies, colony structure and growth behaviour in vitro (distinct growth rates, cell agglomeration, adhesion to culture vessel walls, centrifugation behavior, etc.). All of these aspects are reflected in biomass productivity. Additionally, cell dimensions are also distinct and, as such, the cell density for the same dry weight varies between strains, but this is not considered when expressing the antioxidant yield by dry weight. To normalize these variations, we determined the antioxidant content per cell (or per cellular volume) for each strain. With this normalization, we found that *Nostoc* sp. (LMECYA 291) distinguished itself from all the other strains, with antioxidant levels per cell (or per cellular volume) one to three orders of magnitude higher. 

Indeed, several studies have highlighted *Nostoc* spp. has a promising natural source of antioxidants, comparing with other cyanobacteria [11,21,34]. *Nostoc commune*, for example, is considered a UV-tolerant cyanobacterium [3], because this species produces significant amounts of mycosporine-like amino acids (MAAs), that function as a primary sunscreen [4]. It is known that environmental conditions may influence the variety and quantity of antioxidants in cyanobacteria and that these environmental variations may change daily or seasonally for cyanobacteria in their natural habitats [34]. *Nostoc commume*, as other *Nostoc* spp., is commonly found in terrestrial ecosystems where it is highly exposed to sunlight and subjected to dissection periods. For this reason, these species developed protection strategies such as the production of MAAs and gelatinous matrices formed by polysaccharides that surround *Nostoc* cells preventing dissection [35].

In the present work, we also investigated the biological activity of cyanobacterial extracts. For that purpose, we evaluated the protective effect of the ethanolic extracts of the eight cyanobacteria strains against H_2_O_2_-induced cytotoxicity in the HEK293T human cell line by the MTT assay. This human cell line is commonly used as a model for the investigation of H_2_O_2_ -induced oxidative stress, being often used in studies of the antioxidant activity of natural compounds [36,37]. Pre-exposure of these cells to the majority of the cyanobacterial extracts protected them from the oxidative effect of H_2_O_2_. Survival was higher (between 7% and 52%) in cells pre-exposed to the extracts for 1 h and treated with 0.1 mM H_2_O_2_ for 23 h than in cells pre-exposed to the extracts for 23 h and treated with 1 mM H_2_O_2_ for 3 h (between 5% and 30%). That is, the protective effect of the extracts against the oxidative effect of H_2_O_2_ depends on the time of exposure to the extracts and/or the concentration of the oxidizing agent.

To our knowledge, there are no reported data on the protective effect of cyanobacterial extracts against H_2_O_2_-induced oxidative stress in human cell lines, as has already been described for other natural compounds such as ferrulic acid, a natural phenolic present in plants and fruits [36] and *Phyllanthus phillyreifolius*, an endogenous pant used in traditional medicine in Réunion island [37]. However, a study by Badr et al. [38] revealed a decrease in oxidative stress markers and an increase in endogenous antioxidant defenses in rats under H_2_O_2_-induced oxidative stress, when orally consuming an extract of the cyanobacterial strain *Sphaerospermopsis aphanizomenoides*. In the present study, the extracts of *Nostoc* sp. (LMECYA 291), *Planktothrix mougeotii* (LEGE06224), *Planktothrix agardhii* (LMECYA 257) and *Leptolyngbya* sp. (LMECYA 173) conferred higher protection. These extracts have not exhibited the highest phenolic and flavonoid content. This suggests that other type of compounds present in this extract may be involved in the observed protective activity in HEK293T cells.

## 4. Conclusions

In this work, we demonstrate the antioxidant potential of several cyanobacterial strains from distinct habitats. Overall, freshwater *Aphanizomenon gracile* (LMECYA 009), *Aphanizomenon flos-aquae* (LMECYA 088), terrestrial *Nostoc* sp. (LMECYA 291) and wastewater *Planktothrix mougeotii* (LEGE 06224) seem promising for future studies on biotechnological applications in the field of antioxidants, namely in food, cosmetic and pharmaceutical areas. Further studies would be important to identify the mechanisms underlying the antioxidant activity of these strains and to characterize chemically their antioxidant compounds. 

## 5. Materials and Methods 

### 5.1. Chemicals and Reagents

Absolute ethanol, methanol and chloroform (analytical grade), Folin & Ciocalteu’s phenol reagent, sodium carbonate anhydrous, aluminum chloride, sodium hydroxide and Foetal Bovine Serum (FBS) were purchased from Merck (Darmstadt, Germany). 2,2-diphenyl-1-picrylhydrazyl (DPPH), 6-hydroxy-2,5,7,8-tetramethylchroman-2-carboxylic acid (Trolox) (purity 97%), β-carotene (purity 97.0%), linoleic acid (purity 97.0%), Tween^®^40, (-)-epigallocatechin gallate (purity ≥ 95%), sodium nitrite, quercetin (purity ≥ 95%), Dulbecco’s Modified Eagle’s Medium (DMEM) and 3-(4,5-Dimethyl-2-thizaolyl)-2,5-diphenyl-2H-tetrazolium bromide (MTT) assay were purchased from Sigma-Aldrich (Madrid, Spain). Glucose, Glutamax^TM^ and Trypsin were purchased from Gibco-ThermoFisher Scientific (Dublin, Ireland). Plasmocin™ was purchased InvivoGen (Toulouse, France). 

### 5.2. Cyanobacterial Strains Maintenance and Biomass Production

In this work, six strains isolated from Portuguese surface freshwater reservoirs, one strain isolated from a soil sample and one strain from a wastewater treatment plant (WWTP) were studied.

The freshwater/soil strains belong to the “Estela Sousa e Silva Algae Culture Collection” (ESSACC) and the WWTP strain belongs to the “Blue Biotechnology and Ecotoxicology Culture Collection” (LEGE), as shown in Table 1. These strains were maintained in a culture chamber at 20 °C, with light intensity of 16 ± 4 μEm^−2^s^−1^ and under a 14/10 h light/dark cycle in Z8 medium [39]. Large-scale cultures (5 L) of each strain were prepared in the same conditions and with air-bubbling to ensure nutrients and air mixing and avoid cell deposition. After reaching the exponential growth phase, cyanobacteria biomass was collected by decantation and centrifugation (9600× *g*, 5 min., 4 °C; Beckman J2-14 M/E, Newton, CT, USA). The resulting cellular pellets were freeze-dried (Thermo Scientific Savant Speedvac AES 1000, Waltham, MA, USA) before the extraction of cyanobacterial compounds. Before decantation, a sample of 10 mL was taken for cell counting using Sedgwick-Rafter chambers under an inverted microscope [40]. This sample was also used for measurement of cell dimensions under an optical microscope BX60 coupled with a CCD camera DP11 (Olympus, Tokyo, Japan).

### 5.3. Preparation of Cyanobacterial Extracts 

Cyanobacterial biomass was submitted to a solid-liquid extraction in two distinct solvents: absolute ethanol and methanol. Briefly, three sub-samples of lyophilized biomass (0.5 g) from each cyanobacteria strain were mixed with 10 mL of each of the solvents and stirred at 350 rpm for 30 min. at room temperature, in a horizontal compact stirrer (Edmund Bühler GmbH_KS-15, Hechingen, Germany). The suspensions were centrifuged for 10 min at 5160× *g* in a Thermo Scientific centrifuge (Heraeus Multifuge X3, Altrincham, England). The supernatant was transferred to a pyriform flask and the solvent was totally evaporated in a rotary evaporator (Büchi_R-210, Labortechnik, Switzerland) at 35 °C. The resulting dry residue was weighted, diluted with the corresponding solvent to a final concentration of 1 mg extract DW/mL and stored at −80 °C.

### 5.4. Evaluation of Antioxidant Potential of Cyanobacterial Extract by Chemical Methods

The antioxidant activity of methanolic and ethanolic cyanobacterial extracts was evaluated by the DPPH radical scavenging method and by the β-carotene bleaching assay. The total phenolic compounds and flavonoid were also determined in those extracts. All procedures were performed in triplicate.

#### 5.4.1. DPPH Scavenging Method

The reduction of DPPH radical by cyanobacterial extracts was evaluated according to the method described in Andrade et al. [41]. Briefly, 50 μL of each cyanobacterial extract was added to a 2 mL methanolic solution of DPPH at 14.6 μg/mL. The blank assay consisted of 50 μL of the corresponding solvent extract. Three replicates of each sample extract and blank were prepared. The samples were incubated for 30 min in the dark at room temperature. The absorbance was measured at 515 nm in a spectrophotometer X1_Evolution_300 (Thermo Scientific, Altrincham, England). Radical scavenging capacity was expressed as an inhibition percentage (% IP) of DPPH and calculated using the Equation (1):(%) IP = [(Ac − As30)/Ac] × 100(1)
where: (%) IP―Inhibition percentage of DPPH radical; Ac―Absorbance of the blank; As30―Absorbance of the sample after 30 min. of reaction.

The antioxidant capacity was expressed in µg Trolox equivalents per mL of extract (µg TE/mL) [41,42]. For this purpose, a calibration curve of Trolox was prepared (y = 0.5439x − 0.5861; *r*^2^ = 0.9999), using standard methanolic solution of Trolox, ranging from 10 to 175 μg/mL. 

#### 5.4.2. β-carotene Bleaching Assay

This assay was performed according to Andrade et al. [41]. The method is based on the ability of the different extracts to decrease oxidative losses of β-carotene in a β-carotene/linoleic acid emulsion. A solution of β-carotene at a concentration of 0.2 mg/mL was prepared by dissolving 2 mg of β-carotene in 10 mL of chloroform. In a pear-shaped flask, 200 mg of Tween^®^ 40, 20 mg of linoleic acid and 1 mL of the β-carotene solution were added. Chloroform was evaporated at 40 °C in a rotary evaporator RE-111 with a vacuum pump V-700 from BÜCHI (Flawil, Switzerland). Then, 50 mL of Milli-Q™ distilled water were added and the mixture was vigorously shaken to form an emulsion. Then, 0.2 mL of cyanobacterial extracts were added to 5 mL of the β-carotene emulsion. The samples were submitted to 50 °C for 2 h, in a heating block QBD2 of Grant Instruments (Cambridge, England). For the blank assays, 0.2 mL of the corresponding extract solvent was used. The absorbance of the blank assays was measured at 470 nm before and after 2 h of reaction (at 50 °C) and the absorbance of the samples was measured only after 2 h at the same wavelength. The Antioxidant Activity Coefficient (AAC) was calculated by the Equation (2): AAC = (AS120 − AC120)/(AC0 − AC120) × 1000(2)
where: AAC―Antioxidant Activity Coefficient; AAS120―Absorbance of the sample after 120 min. of reaction; AC120―Absorbance of the blank after 120 min. of reaction; AC0―Absorbance of the blank at the initial time.

#### 5.4.3. Determination of Total Phenolic Compounds

The method was adapted from Erkan et al. [43]. The assay is based on the reduction of the Folin-Cioucalteau reagent by the putative antioxidant compounds of the samples. Briefly, 1 mL of cyanobacterial extracts was added to 7.5 mL of Folin-Cioucalteau reagent (1:10, *v*/*v*) and maintained at room temperature. After 5 min, 7.5 mL of an aqueous sodium carbonate solution (6 mg/mL) was added and the solution was kept in the dark, at room temperature for 2 h. The absorbance was measured at 725 nm. The total content in phenolic compounds were calculated based on the calibration curve of Gallic acid (range: 0.025 to 0.2 mg/mL; y = 6.3518x + 0.112, *r*^2^ = 0.9992) and expressed as Gallic acid equivalents (GAE), in milligrams per gram of the extract (mg GAE/g DW).

#### 5.4.4. Determination of Total Flavonoid Compounds

The total flavonoid content of the extracts was determined according to the method described by Yoo et al. [44]. One mL of the extract was mixed with 4 mL of distilled water and 0.3 mL of a 5% sodium nitrite solution. After 5 min, 0.6 mL of 10% aluminum chloride solution was added. Reaction proceeded for 6 min and, 0.2 mL of 1 M sodium hydroxide and 2.3 mL of distilled water was added. The absorbance of the mixture was recorded at 510 nm. The total flavonoid contents were calculated based on the calibration curve of quercetin (range: 0.1 to 0.7 mg/mL; y = 0.8561x + 0.0304, *r*^2^ = 0.9971) and expressed as quercetin equivalents (QE), in milligrams per gram of the extract (mg QE/g DW).

The results expressed in µg TE/mL (item 5.4.1), mg GAE/g DW (item 5.4.3) and mg QE/g DW (item 5.4.4) were subsequently converted to a “per cell” and to a “per cell volume (µm^3^)” basis by dividing the values of each parameter by the corresponding cell density or cell volume, determined at the time of culture sampling, respectively.

### 5.5. Evaluation of the Protective Effect of Cyanobacterial Extracts against Hydrogen Peroxide Cytotoxicity in HEK293T Cell Line

#### 5.5.1. HEK293T Cell Line Maintenance and Exposure to Cyanobacteria Extracts and Hydrogen Peroxide

The human embryonic kidney HEK293T cell line was used to test the protective effect of cyanobacterial ethanolic extracts against the H_2_O_2_-induced cytotoxicity. Cells were grown in Dulbecco’s Modified Eagle’s Medium (DMEM) supplemented with Foetal Bovine Serum (FBS, 10%), glucose (4.5 g/L), Glutamax™ and Plasmocin™ in a 5% CO_2_ humidified incubator at 37 °C. Adherent cells in exponential growth phase were trypsinized (trypsin, 0.5%), resuspended in culture medium and the cell density and viability were determined by the trypan blue dye exclusion method [45]. Six thousand viable cells were seeded per individual 96-microplate wells and maintained for 24 h for cell adherence and growth, prior to extract exposure. After the 24 h incubation period, the growth medium was removed and cells were exposed to cyanobacterial extracts for a short (1 h) and a long (23 h) period, prior to H_2_O_2_ addition (Figure 8). The extracts were diluted in culture medium at 10 µg biomass/mL and the corresponding ethanol (absolute, anhydrous) concentration was 1%. After the short exposure to the extracts, H_2_O_2_ (0.1 mM final concentration) was added to microplate wells and cells were incubated for additional 23 h. After the long exposure to cyanobacterial extracts, cells were exposed to H_2_O_2_ (1 mM, final concentration) for 3 h. In both conditions, the extracts were co-incubated with H_2_O_2_ (Figure 8). In parallel, cells were exposed to H_2_O_2_ (0.1 mM/23 h or 1 mM/3 h) prepared in fresh DMEM medium but after removing the previous extract-containing medium (Figure 8). Cell exposed only to cyanobacterial extracts (microplate wells 1–3) represents the negative control of cells exposed to cyanobacterial extracts with H_2_O_2_ (microplate wells 4–9). Several additional controls were included in the assay: (a) negative control = culture medium (to confirm the basal cell growth); (b) solvent control = culture medium + ethanol absolute anhydrous at 1% (to ensure that EtOH did not affect cell viability); (c) positive control 1 = H_2_O_2_ (0.1 mM or 1 mM) in culture medium (to ensure H_2_O_2_ impaired cell viability); (d) positive control 2 = H_2_O_2_ (0.1 mM or 1 mM) in culture medium + EtOH (to demonstrate that EtOH did not affect H_2_O_2_ effect). All the exposure conditions were tested in triplicate. Exposure conditions are outlined in Figure 8. 

#### 5.5.2. MTT Assay

The cell viability was evaluated by the 3-(4,5-Dimethyl-2-thizaolyl)-2,5-diphenyl-2H-tetrazolium bromide (MTT) assay [46]. Briefly, after the exposure period to cyanobacterial extracts and H_2_O_2_, the culture medium was removed from microplates and the cells were incubated for 3 h at 37 ^°^C with fresh culture medium containing 10% of MTT solution (5 mg/mL, in PBS). To dissolve the MTT–formazan crystals formed by healthy cells, the MTT containing medium was discharged and a solution of acidified propan-2-ol solution (0.04 M HCl) was added for 15 min under shaking. The absorbance was recorded at 570 nm using a Multiscan Ascent spectrophotometer (Labsystems, Helsinki, Finland).

Cell viability (%) was determined by dividing the mean value of absorbance of each treatment by the mean value of the corresponding control (×100). The viability results of cells exposed to cyanobacterial extracts + H_2_O_2_, with or without a wash step (Figure 6 and Figure 7) were compared with the viability of cells exposed only to cyanobacterial extracts (Figure 6 and Figure 7). In parallel, the additional 4 controls were compared with each other, as represented by the asterisks in Figure 6 and Figure 7.

Cell survival was calculated by subtracting the viability of positive control 1 (culture medium + H_2_O_2_) or positive control 2 (culture medium + EtOH + H_2_O_2_) to the viability of cells exposed to cyanobacterial extract + H_2_O_2_, with or without a previous extract-wash step, respectively.

### 5.6. Statistical Analysis

The statistical analysis of data was done using the IBM SPSS statistics 25 software. The data comply with the normality. Data from the DPPH inhibition assay, β-carotene bleaching test, total phenolic and total flavonoid compounds was examined by One Way ANOVA using Tukey’s (HSD) as post-hoc test. The analysis of the data obtained with MTT assay, to evaluate the protective effect of cyanobacterial extracts against H_2_O_2_ cytotoxicity in HEK293T cell line, was performed by Student’s *t*-test. Differences were considered significant when *p* < 0.05. 

## Figures and Tables

**Figure 1 toxins-12-00548-f001:**
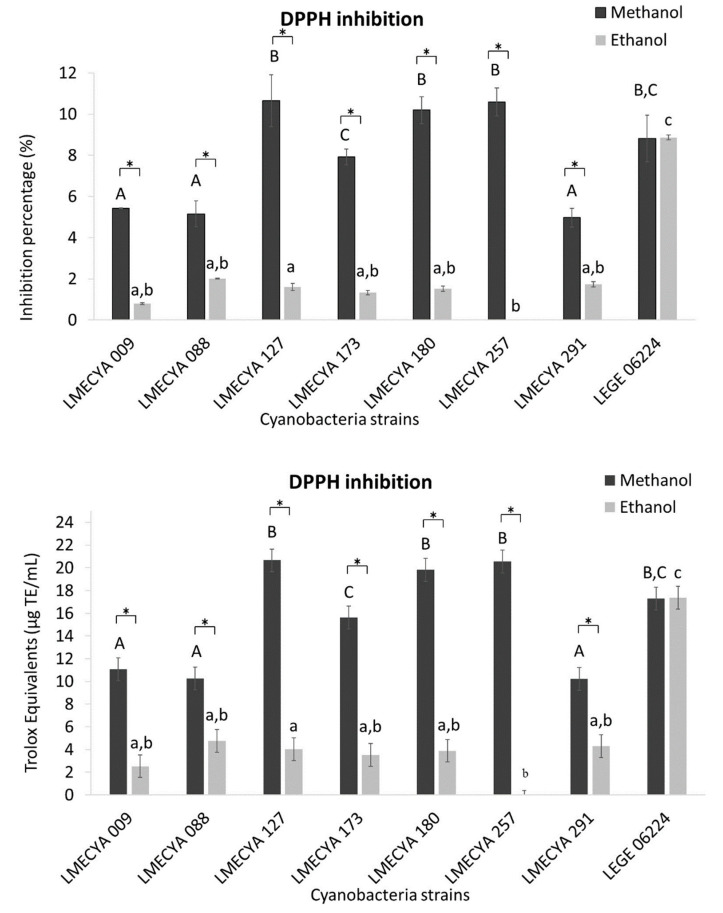
Antioxidant activity of methanolic and ethanolic cyanobacterial extracts (1 mg DW/mL) evaluated by the DPPH inhibition assay and expressed as Inhibition percentage (upper graph) and Trolox equivalents (lower graph). Cyanobacterial strains: *Aphanizomenon gracile* (LMECYA 009), *Aphanizomenon flos-aquae* (LMECYA 088), *Microcystis aeruginosa* (LMECYA 127), *Leptolyngbya* sp. (LMECYA 173), *Dolichospermum flos-aquae* (LMECYA 180), *Planktothrix agardhii* (LMECYA 257), *Nostoc* sp. (LMECYA 291), *Planktothrix mougeotii* (LEGE 06224). * represents a statistically significant difference (*p* < 0.05) between methanolic and ethanolic extracts of each cyanobacterial strain. Different lower-case letters represent significant statistic differences (*p* < 0.05) between the ethanolic extracts of the eight cyanobacteria strains. Different capital letters represent significant statistic differences (*p* < 0.05) between the methanolic extracts of the eight cyanobacteria strains.

**Figure 2 toxins-12-00548-f002:**
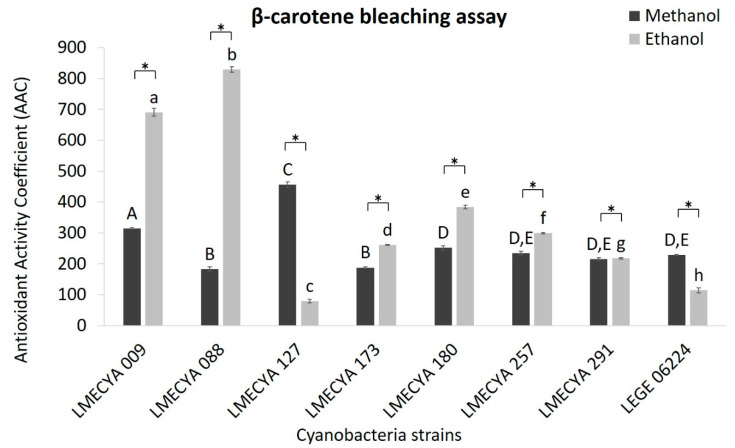
Antioxidant activity coefficient (AAC) of methanolic and ethanolic cyanobacterial extracts (1mg DW/mL) evaluated by the β-carotene bleaching assay. Cyanobacterial strains: *Aphanizomenon gracile* (LMECYA 009), *Aphanizomenon flos-aquae* (LMECYA 088), *Microcystis aeruginosa* (LMECYA 127), *Leptolyngbya* sp. (LMECYA 173), *Dolichospermum flos-aquae* (LMECYA 180), *Planktothrix agardhii* (LMECYA 257), *Nostoc* sp. (LMECYA 291), *Planktothrix mougeotii* (LEGE 06224). * represents a statistically significant difference (*p* < 0.05) between methanolic and ethanolic extracts of each cyanobacterial strain. Different lower-case letters represent significant statistic differences (*p* < 0.05) between the ethanolic extracts of the eight cyanobacteria strains. Different capital letters represent significant statistic differences (*p* < 0.05) between the methanolic extracts of the eight cyanobacteria strains.

**Figure 3 toxins-12-00548-f003:**
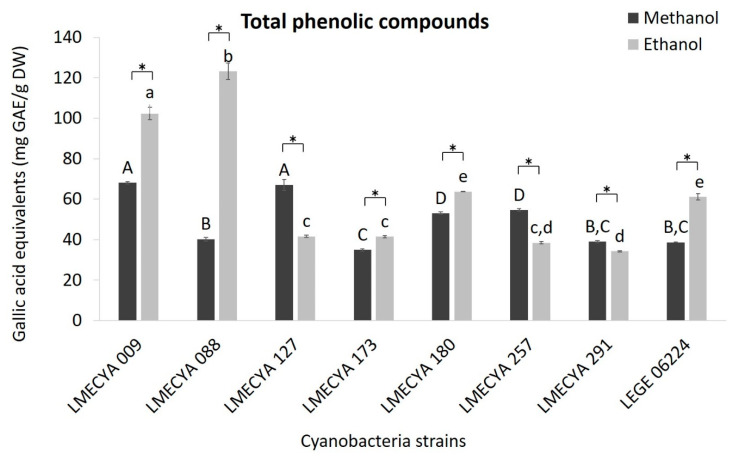
Total phenolic content (mg GAE/g DW) of methanolic and ethanolic cyanobacterial extracts (1 mg DW/mL). Cyanobacterial strains: *Aphanizomenon gracile* (LMECYA 009), *Aphanizomenon flos-aquae* (LMECYA 088), *Microcystis aeruginosa* (LMECYA 127), *Leptolyngbya* sp. (LMECYA 173), *Dolichospermum flos-aquae* (LMECYA 180), *Planktothrix agardhii* (LMECYA 257), *Nostoc* sp. (LMECYA 291), *Planktothrix mougeotii* (LEGE 06224). * represents a statistically significant difference (*p* < 0.05) between methanolic and ethanolic extracts of each cyanobacterial strain. Different lower-case letters represent significant statistic differences (*p* < 0.05) between the ethanolic extracts of the eight cyanobacteria strains. Different capital letters represent significant statistic differences (*p* < 0.05) between the methanolic extracts of the eight cyanobacteria strains.

**Figure 4 toxins-12-00548-f004:**
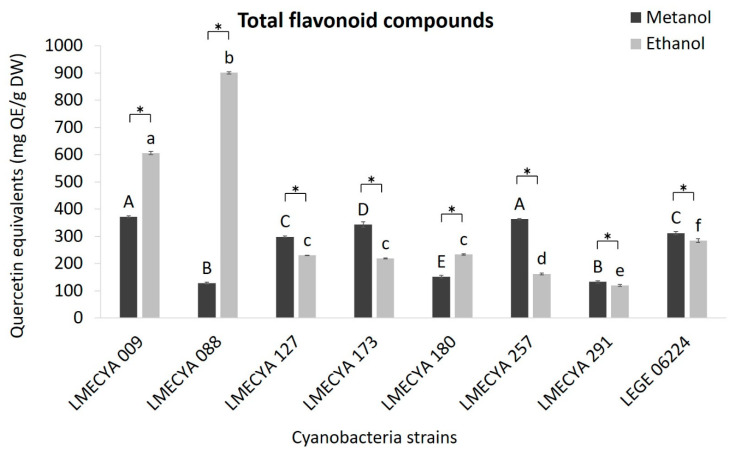
Total flavonoid content (mg QE/g DW) of methanolic and ethanolic cyanobacterial extracts (1 mg DW/mL). Cyanobacterial strains: *Aphanizomenon gracile* (LMECYA 009), *Aphanizomenon flos-aquae* (LMECYA 088), *Microcystis aeruginosa* (LMECYA 127), *Leptolyngbya* sp. (LMECYA 173), *Dolichospermum flos-aquae* (LMECYA 180), *Planktothrix agardhii* (LMECYA 257), *Nostoc* sp. (LMECYA 291), *Planktothrix mougeotii* (LEGE 06224). * represents a statistically significant difference (*p* < 0.05) between methanolic and ethanolic extracts of each cyanobacterial strain. Different lower-case letters represent significant statistic differences (*p* < 0.05) between the ethanolic extracts of the eight cyanobacteria strains. Different capital letters represent significant statistic differences (*p* < 0.05) between the methanolic extracts of the eight cyanobacteria strains.

**Figure 5 toxins-12-00548-f005:**
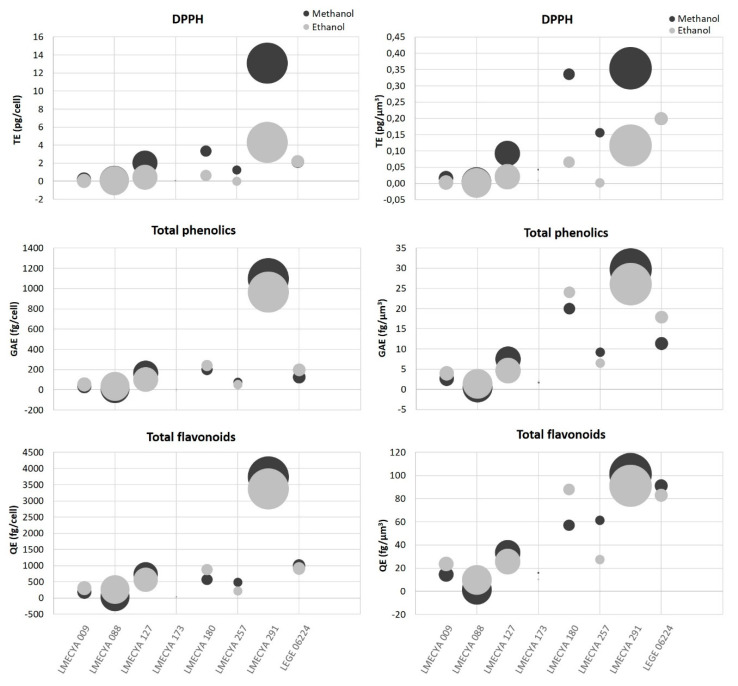
Antioxidant content (DPPH, total phenolic and total flavonoid compounds) of methanolic and ethanolic cyanobacterial extracts per cell (left) or per cell volume (µm^3^) (right). Cyanobacterial strains: *Aphanizomenon gracile* (LMECYA 009), *Aphanizomenon flos-aquae* (LMECYA 088), *Microcystis aeruginosa* (LMECYA 127), *Leptolyngbya* sp. (LMECYA 173), *Dolichospermum flos-aquae* (LMECYA 180), *Planktothrix agardhii* (LMECYA 257), *Nostoc* sp. (LMECYA 291), *Planktothrix mougeotii* (LEGE 06224).

**Figure 6 toxins-12-00548-f006:**
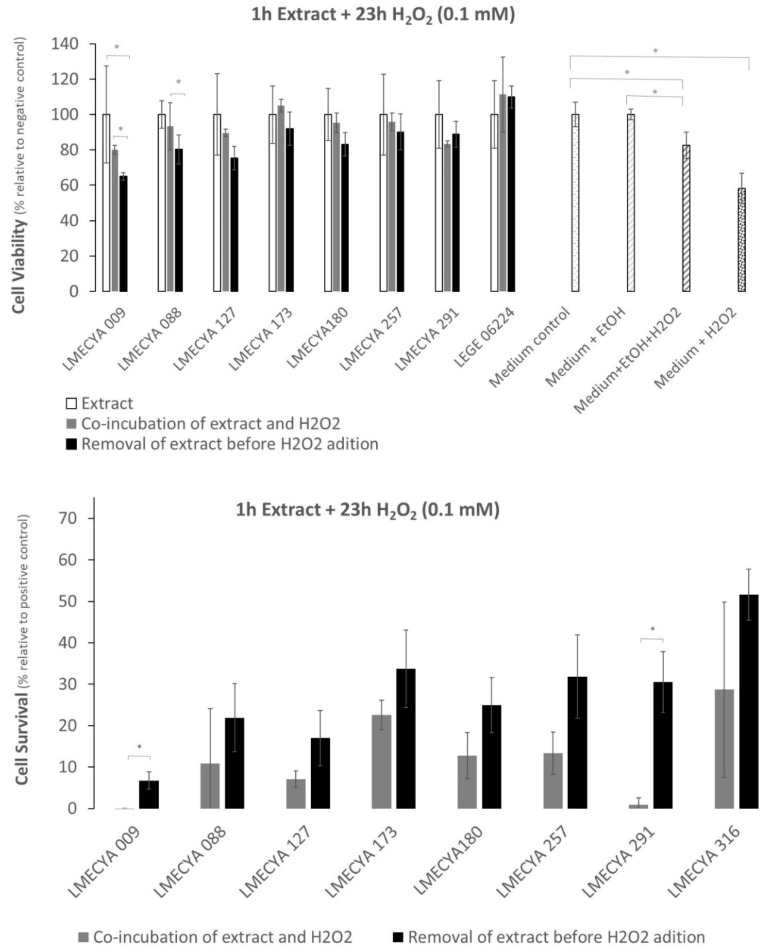
Viability and survival percentage of HEK293T cells exposed to H_2_O_2_ (0.1 mM) for a long period (23 h) and to cyanobacterial extracts for a short period (1 h). Grey bars refer to co-incubation of cells with H_2_O_2_ and cyanobacterial extracts. Black bars correspond to pre-incubation of cells with extracts before H_2_O_2_ exposure. White bars refer to cell exposure to cyanobacterial extracts only. White-dotted bars refer to the distinct controls. * represents a statistically significant difference (*p* < 0.05).

**Figure 7 toxins-12-00548-f007:**
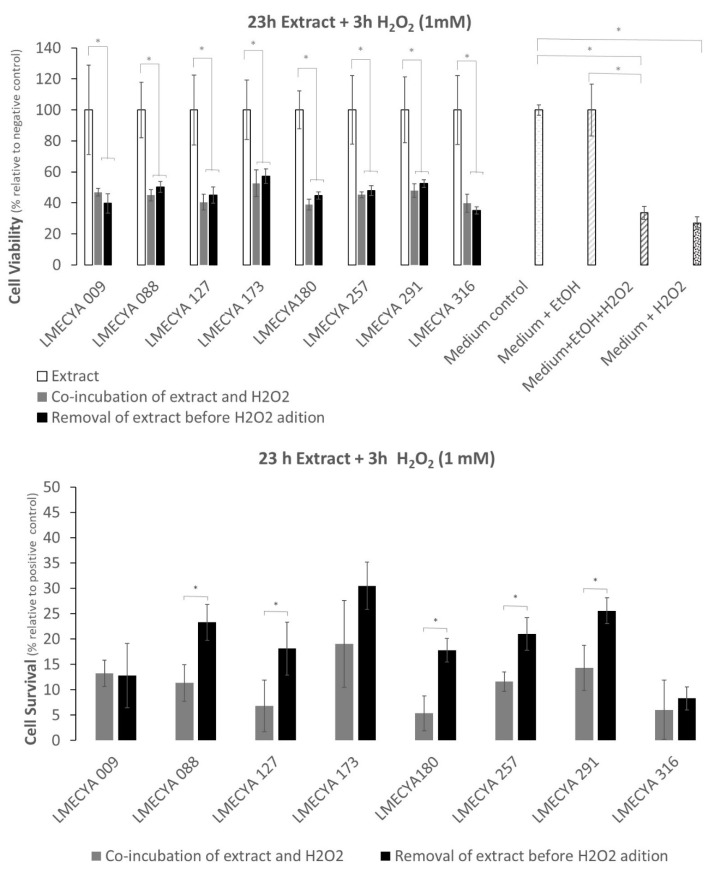
Viability and survival percentage of HEK293T cells exposed to H_2_O_2_ (1 mM) for a short period (3 h) and to cyanobacterial extracts for a long period (23 h). Grey bars refer to co-incubation of cells with H_2_O_2_ and cyanobacterial extracts. Black bars correspond to pre-incubation of cells with extracts before H_2_O_2_ exposure. White bars refer to cell exposure to cyanobacterial extracts only. White-dotted bars refer to the distinct controls. * represents a statistically significant difference (*p* < 0.05).

**Figure 8 toxins-12-00548-f008:**
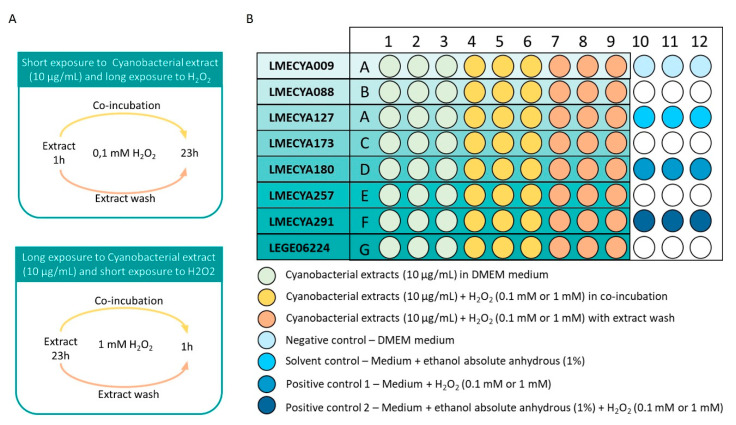
Exposure conditions of HEK293T cells to cyanobacterial extracts (LMECYA and LEGE strains), H_2_O_2_ and respective controls. (**A**) Types of exposure; (**B**) Schematic representation of the 96-microwell plate.

**Table 1 toxins-12-00548-t001:** Cyanobacterial strains, species and origins.

Cyanobacterial Species	Strain Code	Strain Origin
Freshwater strains (ESSACC collection)		
*Dolichospermum flos-aquae*	LMECYA 180	Guadiana River/1999
*Aphanizomenon flos-aquae*	LMECYA 088	Montargil Reservoir/1999
*Aphanizomenon gracile*	LMECYA 009	Peneireiro Reservoir1996
*Leptolyngbya* sp.	LMECYA 173	Hydrothermal fountain /2003
*Microcystis aeruginosa*	LMECYA 127	Montargil reservoir/2000
*Nostoc* sp.	LMECYA 291	Garden soil /2013
*Planktothrix agardhii*	LMECYA 257	São Domingos Reservoir /2009
WWTP strain (LEGE collection)		
*Planktothrix mougeotii*	LEGE 06224	Febros WWTP /2006

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
