# Peer review of "Antioxidant and Cytoprotective Properties of Cyanobacteria: Potential for Biotechnological Applications"

_toxins, 2020, doi:10.3390/toxins12090548_

Round 1

Reviewer 1 Report

Toxins-850244

88, 89: ACC? Probably should be AAC. Please define the abbreviation at first use.

106: GAE not defined.

147: The statement that “we could expect that cells with higher volumes would have higher content on antioxidant compounds” does not make sense because fewer large cells than small cells would be sampled on a per weight basis (a defined dry weight was extracted).

Figure 5: Why change the order of the extracts on this graph? It makes it hard to compare with other graphs.

Figure 6: It is unclear what the comparator was for the statistical tests. It is clear ethanol was protective as Medium+EtOH+H2O2 is higher than H2O2 alone so the EtOH control should be the comparator for identifying significant effects of the extracts, but this is not stated. The data in the Cell Survival graph aren’t correctly calculated. Viability in the upper graph was generally greater than 80% with extract but about 50% in H2O2 alone. Hence in wells where extracts were removed, extracts prevented about 60% of sensitive cells (80%-50%/50%) from dying. However, in most coincubation wells there was little difference in survival if EtOH plus H2O2 is the comparator.

Discussion: this could be shortened considerably by only citing other author’s results if they are actually comparable to the results of this study. It appears that many of the cited results are not directly comparable due to “differences in endpoints and methods used” (line 300).

354: Phenol reagent Folin & Ciocalteu’s should be Folin & Ciocalteu’s phenol reagent

359, 454: Dulbecco's Modified Eagle Medium. Should be Eagle’s

360: Thiazolyl Blue Tetrazolium Blue (MTT). Not the standard name for MTT and certainly don’t need 2 Blues. Referred to on line 477 as 3-(4,5-Dimethyl-2- thizaolyl)-2,5-diphenyl-2H tetrazolium bromide (MTT) assay.

360, 362, 373, 374, 385, 401: USA, not EUA.

362: What does “MTT Calbiochem, Darmstadt, Germany” refer to?

Table 1: Hydrothermal not Hydrotermal.

386: Switzerland not Suiss.

394, 411: In both of these absorbance methods, how was the potential for interfering absorbance by the cyanobacterial extracts controlled for? Why wasn’t a positive control included in the β-carotene assay?

432: remove one “the”.

462: Were only the ethanoic extracts tested since only the final concentration of this solvent is mentioned? Please state this clearly.

Figure 8: It appears from this diagram that the order of extracts was the same in each replicate plate. If so, how as edge effect controlled for (A and G having slightly lower absorbance due to having one clear side to the wells). This experiment would have been better done using concentration ranges of the extracts rather than just a single concentration.

485: The Student’s t-test is not appropriate for the multiple comparisons performed in this work. P=0.05 means that there is a 1/20 chance of a comparison being found significant purely by chance. The authors appear to have performed about 30 comparisons for each experiment. The data need to be reanalysed using 2-way ANOVAs with an appropriate post-hoc test to show significant differences from control. The SPSS software used is certainly capable of this.

Author Response

The authors thank reviewer 1 for his/her constructive critics and suggestions that contributed to the improvement of the manuscript.

88, 89: ACC? Probably should be AAC. Please define the abbreviation at first use

Answer: The change was made accordingly. 

106: GAE not defined

Answer: The change was made accordingly. 

147: The statement that “we could expect that cells with higher volumes would have higher content on antioxidant compounds” does not make sense because fewer large cells than small cells would be sampled on a per weight basis (a defined dry weight was extracted). 

Answer: The authors agree with the reviewer. All the paragraph was changed and simplified (lines 145-151/revised version).  A sentence was added in M&M section to explain the conversion of results to cell and to cell volume (lines 523-526/revised version). A sentence was also added in line 433-435 (revised version) to explain how cell measurements were done. 

Figure 5: Why change the order of the extracts on this graph? It makes it hard to compare with other graphs. 

Answer: The order of the extracts was changed because the authors wanted to display the values of the antioxidants vs the increasing volume size of the cyanobacterial strains. However, the authors accepted the reviewer's suggestion and maintained the order of the extracts as shown in the previous figures. Additionally, the authors detected that in the legend of this figure, the terms “upper graphic” and “lower graphic” were incorrect. Those terms were changed to “left” and “right”, respectively.

Reviewer/Figure 6: It is unclear what the comparator was for the statistical tests.  

Answer: To clarify this issue the following paragraph was added at 5.5.2. “The viability results of cells exposed to cyanobacterial extracts+H2O2, with or without a wash step (grey and black bars, respectivelly,  in Figures 6 and 7) were compared with the viability of cells exposed only to cyanobacterial extracts (white bars in Figure 6 and 7). In parallel, the additional 4 controls were compared with each other, as represented by the asterisks in Figures 6 and 7”. Besides, more detailed information about the controls was included in item 5.5.1 (lines 486-491/revised version). Fig 8 was changed accordingly. 

Reviewer/Figure 6: It is clear ethanol was protective as Medium+EtOH+H2O2 is higher than H2O2 alone so the EtOH control should be the comparator for identifying significant effects of the extracts, but this is not stated. 

Answer: Indeed, according to Figure 6 (1h extract/23h H2O2) the EtOH seems to reduce the H2O2 effect. Even though, a significant difference was observed between Medium+EtOH+H2O2 and Medium+EtOH. Please note that in Figure 7 (23h extract/3h H2O2) the EtOH did not influenced the effect of H2O2. But, as referred above, the results of cells exposed to cyanobacterial extracts plus H2O2 were compared with the results of cells exposed only to cyanobacterial extracts.  All cyanobacterial extracts were prepared in EtOH, as described in M&M (5.3).  

Reviewer/Figure 6: The data in the Cell Survival graph aren’t correctly calculated. Viability in the upper graph was generally greater than 80% with extract but about 50% in H2O2 alone. Hence in wells where extracts were removed, extracts prevented about 60% of sensitive cells (80%-50%/50%) from dying. However, in most coincubation wells there was little difference in survival if EtOH plus H2O2 is the comparator. 

Answer: The authors agree with the reviewer. The comparator should be EtOH plus H2O2 (positive control 2, in the revised version) in the case of coincubation. To clarify this issue, the sentence “Cell survival was calculated by subtracting the viability of positive control 1 (culture medium+H2O2) or positive control 2 (culture medium+EtOH+H2O2) to the viability of cells exposed to cyanobacterial extract+H2O2, with or without a previous extract-wash step, respectively” was added to 5.5.2 (lines 513-515/revised version). Results of cell survival were recalculated and the corresponding graphics were corrected. The results (item 2.2) and discussion were also changed, accordingly. 

Discussion: this could be shortened considerably by only citing other author’s results if they are actually comparable to the results of this study. It appears that many of the cited results are not directly comparable due to “differences in endpoints and methods used” (line 300)

Answer: Despite several cited results are not direct comparable with the results presented in this manuscript, as referred in line 303, the authors consider that those citations were important since they give a broad picture of the antioxidant ranges found in cyanobacteria and other organisms.  

354: Phenol reagent Folin & Ciocalteu’s should be Folin & Ciocalteu’s phenol reagent 

Answer: The change was made accordingly.  

359, 454: Dulbecco's Modified Eagle Medium. Should be Eagle’s 

Answer: The change was made accordingly. 

360: Thiazolyl Blue Tetrazolium Blue (MTT). Not the standard name for MTT and certainly don’t need 2 Blues. Referred to on line 477 as 3-(4,5-Dimethyl-2- thizaolyl)-2,5-diphenyl-2H tetrazolium bromide (MTT) assay. 

Answer: The change was made accordingly. 

360, 362, 373, 374, 385, 401: USA, not EUA

Answer: The corrections were made. 

362: What does “MTT Calbiochem, Darmstadt, Germany” refer to? 

Answer: This was a lapse and it was removed. 

Table 1: Hydrothermal not Hydrotermal. 

Answer: The correction was made. 

386: Switzerland not Suiss

Answer: The correction was made. 

394, 411: In both of these absorbance methods, how was the potential for interfering absorbance by the cyanobacterial extracts controlled for? Why wasn’t a positive control included in the β-carotene assay? 

Answer: As referred in section 5.3.2, the antioxidant activity was calculated by the formula 

 ??? = (AS120-AC120) / (AC0-AC120) x 1000, where the AS sands for the absorbance of the samples at the end of the 120 minutes and the AC stands for the absorbance of the control at before and after the exposure to 50 ºC. The reaction is based on the ability of a determined sample to inhibit or delay the oxidation of an emulsion of linoleic acid and β-carotene, by capturing the linoleic acid free radicals. As the β-carotene is released from the complex between it and linoleic acid, the yellow colour of the reaction mixture is regained. Hence stronger the antioxidant present in the reaction mixture brighter will be the colour and higher will be its optical density. For the purpose of this study, the authors did not found any article or reference to the β-carotene Bleaching Assay being performed using positive controls.

432: remove one “the”

Answer: The correction was made. 

462: Were only the ethanoic extracts tested since only the final concentration of this solvent is mentioned? Please state this clearly. 

Answer: Yes, only the ethanolic extracts were tested. This was mentioned in Discussion section (line 326).  In order to clarify, this was added in the M&M and Results (Lines 164 and 469 in the revised version). 

 Figure 8: It appears from this diagram that the order of extracts was the same in each replicate plate. If so, how as edge effect controlled for (A and G having slightly lower absorbance due to having one clear side to the wells). This experiment would have been better done using concentration ranges of the extracts rather than just a single concentration. 

Answer: The authors understand the issue posed by the reviewer and, indeed, the edge effect was not addressed. But if evaporation have occurred in lanes A and G, this would have had repercussion both at the oxidant agent (H2O2) as well as at the cyanobacterial extracts. This might have been minimized since cells were maintained in a humidified incubator. Nevertheless, the results from the negative controls (medium and medium+EtOH) and positive controls (medium+EtOH+H2O2 and medium+ H2O2) were as expected, which ensure that the assay proceeded well. Although the authors agree that testing concentration ranges of extracts would be a better approach, the quantity of extract was a limitation. This study was a first approach to the antioxidant activity of cyanobacterial strains from ESSACC collection. It is our intention to further study the strains that have proved most promising. For that purpose, we have to produce and extract higher amounts of cyanobacterial biomass.

485: The Student’s t-test is not appropriate for the multiple comparisons performed in this work. P=0.05 means that there is a 1/20 chance of a comparison being found significant purely by chance. The authors appear to have performed about 30 comparisons for each experiment. The data need to be reanalysed using 2-way ANOVAs with an appropriate post-hoc test to show significant differences from control. The SPSS software used is certainly capable of this. 

Answer: This is, indeed, a mistake. The statistical analysis was correctly performed but it was not correctly mentioned in the manuscript. The Student’s t-test was used to analyse the results from the MTT assay with HEK293T cells. One Way ANOVA was used to analyse data from the DPPH inhibition assay, β-carotene bleaching test, total phenolic and total flavonoid determination. This was corrected in item 5.6.

Reviewer 2 Report

The manuscript is well written and organized. However, the authors tested the antioxidant activity using non physiological radicals, as in the assay of DPPH, or indirect assays as in β-carotene bleaching assay. Moreover, the use of a cellular model, despite being important for understanding the antioxidant effects of the compounds under study, the authors only tested the viability of the cells using a currently used method, the MTT assay. Therefore in my opinion, the manuscript does not have sufficient scientific sound to be published.

Author Response

The authors do not understand the reviewer´s comments. Why did the reviewer consider that DPPH and β-carotene bleaching assays are not proper methods to evaluate the antioxidant properties of cyanobacteria? The authors fear that the reviewer did not fully understood the purpose of the work. In this study, we intended to evaluate the antioxidant potential of cyanobacterial strains belonging to the ESSACC collection. In a first stage, we used chemical methods to assess the antioxidant activity of extracts obtained from those cyanobacterial strains (DPPH and β-carotene bleaching assays) as well as the total phenolic and flavonoid content of those extracts. These assays are commonly used to evaluate the antioxidant activity of algae and plants and have also been used in studies with cyanobacteria. However, every method has its strengths and weaknesses and the fact that DPPH uses non-physiological radicals is one of the drawbacks. Although some of the long-standing assays like the TPC and DPPH assay suffer from a number of drawbacks, the data gleaned from such assays are still useful as long as they are interpreted and represented correctly [Tan and Lim. Critical analysis of current methods for assessing the in vitro antioxidant and antibacterial activity of plant extracts. Food Chemistry 172 (2015) 814–822, https://doi.org/10.1016/j.foodchem.2014.09.141]. In this sense, the authors consider that the methods employed in this study are adequate for the purpose of an initial screening of the antioxidant activity of cyanobacterial extracts. In a second stage, we tested the hypothesis that those extracts could revert or prevent the cytotoxic effect of an oxidant agent in a biological model. For that purpose, we evaluated the protective effect of the cyanobacterial extracts in HEK293T cells exposed to H2O2. Hydrogen peroxide is a strong oxidant in cells and induces marked decrease in mammalian cells viability. As the authors referred in line 324 "This cell line is one of the models of human cells under oxidative stress induced by H2O2 that is often used in studies of the antioxidant activity of natural compounds [36,37]”. The MTT assay is largely used to determine the viability of mammalian cells. This work constitutes a first approach to evaluate the antioxidant potential of cyanobacterial strains belonging to the ESSACC collection. This study was a preliminary screening to identify and select the most promising strains to use in further studies designed to identify and characterize the specific antioxidant compounds present in the selected strains. 

Round 2

Reviewer 1 Report

The authors appear to have responded adequately to my review comments

Reviewer 2 Report

Dear Authors,

Despite your response to my comments, I appologize, but I mantain all my observations.

The authors tested the antioxidant activity using non physiological radicals, as in the assay of DPPH, or indirect assays as in β-carotene bleaching assay. Almost all the compounds shown a scavenging effect against this type of radicals. Moreover, the use of a cellular model, despite being important for understanding the effects of the compounds under study, the authors only tested the viability of the cells using a current used method, the MTT assay. This is an important but single information.

Therefore in my opinion, the manuscript does not represent sufficient novelty nor results that can be physiologically transposed.